# Application of Nonlinear Adaptive Control in Temperature of Chinese Solar Greenhouses

**Yonggang Wang, Yujin Lu \* and Ruimin Xiao**

College of Information and Electronic Engineering, Shenyang Agricultural University, Shenyang 110866, China; wygvern@syau.edu.cn (Y.W.); xrm@stu.syau.edu.cn (R.X.)
\* Correspondence: lyj@stu.syau.edu.cn; Tel.: +86-151-4215-8410

**Abstract:** The system of a greenhouse is required to ensure a suitable environment for crops growth. In China, the Chinese solar greenhouse plays a crucial role in maintaining a proper microclimate environment. However, the greenhouse system is described with complex dynamic characteristics, such as multi-disturbance, parameter uncertainty, and strong nonlinearity. It is difficult for the conventional control method to deal with the above problems. To address these problems, a dynamic model of Chinese solar greenhouses was developed based on energy conservation laws, and a nonlinear adaptive control strategy combined with a Radial Basis Function neural network was presented to deal with temperature control. In this approach, nonlinear adaptive controller parameters were determined through the generalized minimum variance laws, while unmodeled dynamics were estimated by a Radial Basis Function neural network. The control strategy consisted of a linear adaptive controller, a neural network nonlinear adaptive controller, and a switching mechanism. The research results show that the mean errors were 0.8460 and 0.2967, corresponding to a conventional PID method and the presented nonlinear adaptive scheme, respectively. The standard errors of the conventional PID method and the nonlinear adaptive control strategy were 1.8480 and 1.3342, respectively. The experimental results fully prove that the presented control scheme achieves better control performance, which meets the actual requirements.

**Keywords:** Chinese solar greenhouse; temperature control; nonlinear adaptive control; radial basis function neural network

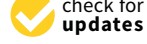



## 1. Introduction

A greenhouse is devised to create a more favorable climate, such as enough solar radiation, adequate temperatures, and suitable humidity, to protect plants and promote the growth of crops [1–3]. Chinese solar greenhouses (CSGs) produce a significant annual output, which has large economic and social benefits in China. Therefore, considerable attention has been given to the CSG in order to provide a proper environment for crop growth. However, in Northeast China, the average temperatures are very low, even falling to below −10 °C, and the cold season generally lasts for four months due to the high latitude in some areas [4]. Such arctic weather seriously affects normal production and brings great loss to the economic benefits [5]. Therefore, the structure of the CSG is different from other countries due to the variability of weather in Northeast China, which usually consists of a south roof, a north roof, a north wall and a thermal blanket for supplying pollution-free and high-quality vegetables even during the winter.

A great many modeling methods for greenhouses have already been proposed such as mechanism, transfer function and black-box modeling [6]. The Takagi-Sugeno fuzzy model was developed according to the historical data of greenhouses [7]. A modeling approach was built using the data collected from an actual greenhouse under closed-loop control [8]. Online identification technology was adopted to obtain a more accurate greenhouse model [9]. A stochastic dynamic model based on parameter identification was proposed through the maximum likelihood estimation method [10]. It should be noted that

dynamic models of greenhouses in different areas have differences owing to construction and covering materials, which directly affect the internal environmental conditions, such as temperature and relative humidity. Therefore, the dynamic model of the CSG is different from other greenhouse because of its special structure.

The inside temperature is a significant factor in restricting greenhouse production in Northeast China [11]. In this situation, the CSG must maintain a certain indoor temperature level to meet the needs of crop growth. Furthermore, determining automatic control strategies is the leading goal for obtaining higher-quantity greenhouse crops. A classical feedback controller, such as the PID control method [12], has been widely used in various fields. However, the control of inside temperatures has generally confronted a series of difficulties in applying the classical feedback control strategy due to its inherent stochastic complexity properties as follows:

(1) The greenhouse is considered a nonlinear dynamic system with intensive multi-disturbance from surroundings, such as global radiation, humidity, and outside air temperature [13,14];

(2) The control process is severely influenced by instable factors including global radiation, external weather, and human activities;

(3) The crops and the environment have a strong and interactive relationship [15]. For example, the plants transpiration and photosynthesis similarly affect the greenhouse temperature that they depend on.

During recent years, many scholars have proposed advanced control strategies, such as adaptive control [16,17], feedforward control [18,19], optimal control [20,21], fuzzy control [22,23], robust control [24,25], and so on. These control methods can ensure the inside temperature near the temperature set point in certain conditions. However,, the problems caused by the instable factors and multi-disturbances are still difficult to deal with. Furthermore, most of these climate control strategies are difficult to carry out in greenhouse production due to the theoretical complex.

Deterministic artificial intelligence can deal with deterministic self-awareness statements based on either the physics of the underlying problem or system identification to establish governing deferential equations [26]. Furthermore, stochastic artificial intelligence, such as neural network technologies, can be expressed in stochastic algorithms in the face of stochastic disturbances. Adaptive controller design of a nonlinear system with discrete-time characteristics was studied using neural networks [27]. The stochastic neural adaptive tracking control problem of an indeterminate switched nonlinear system with a non-strict feedback characteristic was investigated in [28]. The adaptive neural network control scheme was presented to solve the accurate and robust control problem of nonlinear systems with unknown dynamics [29]. The adaptive neural network controller, based on the technique of backstepping, was proposed for the consensus tracking control problem [30]. A nonlinear adaptive decoupling switching control strategy using neural networks was studied to improve the closed-loop performance and evaporation efficiency [31].

The above considerations motivated our study. In particular, inspired by the modeling method and control method in [32–34], this paper starts with the development of a dynamic model of the CSG and proposes a nonlinear adaptive control scheme based on Radial Basis Function (RBF) neural networks to solve temperature control for the CSG system. The main contributions of this paper are summarized as follows:

(1) In order to make the dynamic model more accurate and closer to the actual system, heat transfer quantities of the north wall and north roof were respectively added to the dynamic model in this paper. In addition, the cold air penetration was added to the humidity balance model;

(2) To the best of our knowledge, almost no research so far has addressed the fact that the existing control scheme based on RBF has been applied to the CSG considering the nonlinearity and adaptiveness. This control approach takes advantage of the strong ability of learning and adaptability of RBF neural networks. In this paper, a linear

adaptive controller, a neural network nonlinear adaptive controller, and switching mechanism were combined to improve dynamic performance on the promise of guaranteeing system stability. The parameters of the controller were determined based on the generalized minimum variance control law. An RBF neural network was employed to solve the unmodeled dynamics of CSGs. The experimental results express that the presented control strategy shows quick set-point tracking ability in the case of multi-disturbances and can achieve satisfactory control performances.

## 2. Materials and Methods

### 2.1. CSG Facility

The research on modeling and control was performed in an experimental greenhouse located at Shenyang Agricultural University in Liaoning Province (41.48° N, 123.24° E, and 42 m a.s.l. (above sea level)). The CSG architecture schematic is shown in Figure 1. The greenhouse was 60 m long and 12 m wide. The heights of the north wall and north roof were 3 m and 5.5 m, respectively. The north wall was a 0.6 m thick layered structure consisting of brick, insulation polystyrene foam, and air layer. The cover of the south roof was made of a 0.00012 m thick polyvinyl chloride film, and a 0.02 m thick cotton blanket was used for thermal insulation at night.

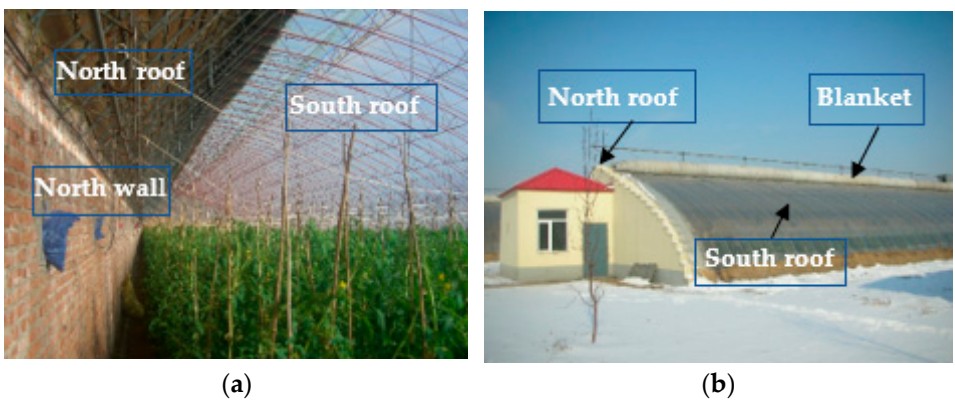

|      (**a**)      |      (**b**)      |

**Figure 1.** Architecture schematic of the (**a**) inside, and (**b**) outside of a CSG.

For measurement of the ambient parameters, different sensors were installed in the experimental greenhouse. The type of temperature and humidity sensor used was the SHT10 (Sensirion, Zurich, Switzerland), which can measure the temperature and humidity simultaneously. Eighteen sensors were installed in the experimental greenhouse. A distribution diagram of environmental sensors is shown in Figure 2. The locations of sensors were installed every 18 m and two sets of sensors were installed at each location. The sensors were placed horizontally at different heights above the ground (1.5, 3, and 4.5 m). The mean of the temperature and humidity recorded by the eighteen installed sensors was considered as the temperature inside the greenhouse. The inside solar radiation and outside solar radiation were measured by a pyranometer (MP200, Apogee Instruments, Logan, UT, USA). The outside temperature, outside humidity, and the direction and velocity of the wind were collected from an outer weather station. The sample interval of parameters was 5 s and the mean of every 900 s was recorded for all variables.

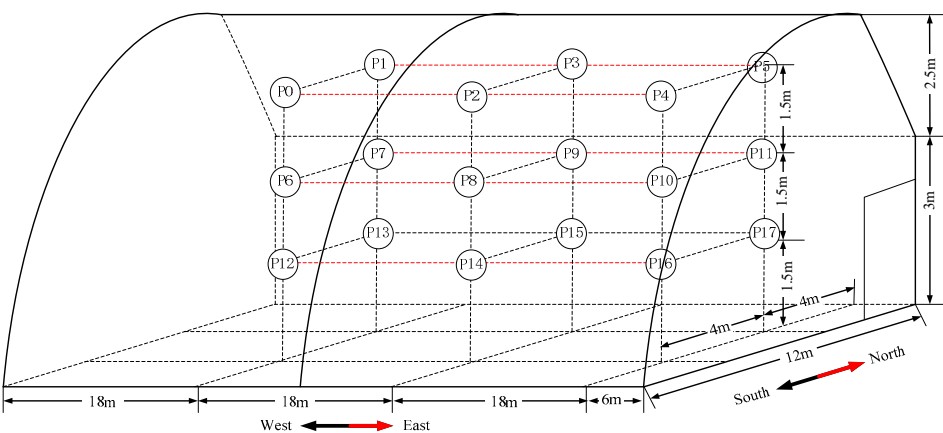

**Figure 2.** Distribution diagram of environmental sensors.

The Shenyang district (in Northeast China) is classified as a monsoon climate, and has a wide temperature range. Influenced by the monsoon, winters are long, lasting nearly five months. Solar radiation is able to provide part of the heat for plant growth. However, during the night or bad weather such as snow with insufficient light intensity, the greenhouse plays a crucial role in maintaining an appropriate temperature in order to prevent the plants from damage, which would markedly decrease their production rate. Therefore, it is common to regulate the temperature in these areas during the winter, especially for value-added crops, such as strawberry or mushroom.

### 2.2. Greenhouse Model Description

The greenhouse dynamic model is usually obtained according to the energy conservation principle [35]. Considering the characteristics of the unique structure in CSGs, heat transfer quantity $Q_w \left( \text{W/m}^2 \right)$ and $Q_m \left( \text{W/m}^2 \right)$ are introduced from inside air to the north wall and north roof, respectively [36]. The greenhouse model was developed in this paper according to energy balance (Figure 3). These differential equations are given by:

$$\begin{cases} \frac{dT_{in}}{dt} = \frac{Q_{rad} + Q_{heat} - Q_c - Q_n - Q_s - Q_m - Q_w - Q_r}{\rho C_P h} \\ \frac{dH_{in}}{dt} = \frac{E - C - \phi_a - \phi_e}{h} \end{cases} \tag{1}$$

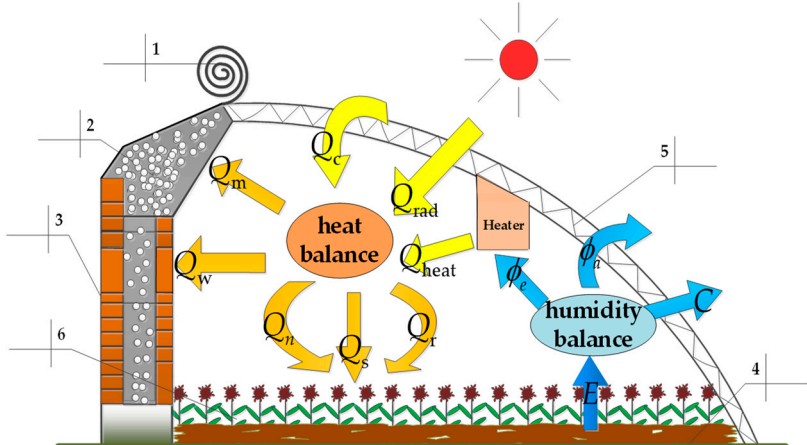

**Figure 3.** Schematic diagram of energy balance of experimental greenhouse. (Note: 1. the greenhouse blanket; 2. the north roof; 3. the north wall; 4. the soil; 5. the greenhouse frame; 6. crops).

In the heat model of Equation (1), where $\rho$ is the air density, $C_P$ is air specific heat capacity, $h$ is the height of the greenhouse, $Q_{rad}$ $\left( \text{W/m}^2 \right)$ is the intercepted solar radiant

energy, $Q_{heat}$(W/m$^2$) is the heat provided by the greenhouse heaters, $Q_c$(W/m$^2$) is the heat transferred from the envelope between the outside and the inside, $Q_r$(W/m$^2$) is the heat absorbed by the crops through transpiration, $Q_n$(W/m$^2$) is the sensible heat transferred from inside air to crops, $Q_s$ (W/m$^2$) is the heat transferred from inside air to the soil in the greenhouse, $Q_w$(W/m$^2$) is the sensible heat transferred from the north wall to indoor air, and $Q_m$(W/m$^2$) is the sensible heat transferred from north roof to inside air.

In the humidity model of Equation (1), where $E$ is the transpiration rate of crops in g·m$^{-2}$·s$^{-1}$, $C$ is the water vapor condensation caused by the indoor and outdoor temperature difference in g·m$^{-2}$·s$^{-1}$, $\phi_a$ is the humidity taken away by the cold air penetrating the greenhouses in g·m$^{-2}$·s$^{-1}$ and $\phi_e$ is the water condensation or evaporation when heating system is activated in g·m$^{-2}$·s$^{-1}$.

According to the well-known Penman–Monteith formula, $Q_r$ can be circulated by [37]:

$$Q_r = C_l R_n + \left(\frac{e_s \beta}{\gamma}\right) T_{in} - \left(\frac{h_l P}{8.036\gamma}\right) H_{in} \tag{2}$$

where $C_l$ is the convective heat loss coefficient from indoor air to the cover, $R_n$ is the net radiative exchange between the canopy and the environment, $e_s$ is the air saturation vapor pressure, $\beta$ is the influence coefficient of temperature change on saturated water vapor pressure, $\gamma$ is the psychrometric constant, $h_l$ is the heat transfer constant between crops and inside air, and $P$ is the standard atmospheric pressure. $H_{in}$ indicates the absolute humidity of the indoor air.

Solar radiation, a significant factor affecting the indoor temperature, is defined as [38]:

$$Q_{rad} = \frac{c_1 \tau S_{out} A_{gr} sin I^o}{V_a} \tag{3}$$

where $c_1$ is the aging coefficient of lighting material, $\tau$ is the greenhouse global transmission, $S_{out}$ is the solar radiation, $A_{gr}$ is surface area of greenhouse which absorbs solar radiation, $sin I^o$ is the incidence angle of sunlight, and $V_a$ is the volume of the greenhouse.

In this study, the air heating adopted by heating equipment, the energy provided by which is calculated as [39]:

$$Q_{heat} = \frac{\eta Q_P}{A_{gr}} \tag{4}$$

where $\eta$ is the energy efficiency of heaters and $Q_P$ is the energy power of the heating equipment in W/m$^2$.

$Q_c$ can be expressed as follows [40]:

$$Q_c = h_C (t_{in} - t_{out}) \left(\frac{A_{su}}{A_{gr}}\right) \tag{5}$$

where $t_{in}$ and $t_{out}$ are the inside and outside temperatures in °C, respectively. $A_{su}$ is the superficial area of cover materials. The conversion relation between $t$ and $T$ is as follows:

$$t = T - 273.15 \tag{6}$$

The overall energy loss coefficient $h_c$, increasing with wind speed $v_{out}$, is defined as the formula [38]:

$$h_c = A + B v_{out} \tag{7}$$

Single-layer and double-layer covering materials are different. Generally, CSGs use a single-layer covering material. In this paper, the values of $A$ and $B$ were 6 and 0.5, respectively.

In this paper, the heat transfer into the greenhouse environment was also considered. In Equation (1), $Q_n$ and $Q_s$ are calculated as follows [13]:

$$Q_n = (T_{in} - T_l)\rho C_P / r_a \tag{8}$$

$$Q_s = (T_{in} - T_s)\rho C_P / r_a \tag{9}$$

where: $T_s$ is soil surface temperature and $T_l$ represents the leaf temperature of crops.

Under existing conditions in the CSG, this paper calculated the aerodynamic resistance of soil by means of following formula [37]:

$$r_a = 305(D/v_{in})^{0.5} \tag{10}$$

where: $D$ is the leaf width and $v_{in}$ is indoor wind speed.

Heat transfer quantity $Q_w$ is calculated as follows:

$$Q_w = A_w \alpha_w (T_{in} - T_w) \tag{11}$$

where: $A_w$ is the north wall area, $T_w$ represents the north wall temperature and $\alpha_w$ is the convective heat transfer coefficient between the north wall and the inside air.

Heat transfer quantity $Q_m$ is calculated as follows:

$$Q_m = A_h \alpha_h (T_{in} - T_h) \tag{12}$$

in which $A_h$ is the area of the north roof, $T_h$ represents the north roof temperature and $\alpha_h$ is the convective heat transfer coefficient between the north roof and the inside air.

$$E = \frac{C_l R_n + (e_s \beta/\gamma) T_{in} - (h_l P/8.036\gamma) H_{in}}{\lambda} \tag{13}$$

where $\lambda$ is latent heat of evaporation.

$$C = 0.00164 \left( A_r / A_{gr} \right) \left( T'_{in} - T'_r \right)^{1/3} (H_{in} - H_{s,r}) \tag{14}$$

in which, $A_r$ is the greenhouse covering the area in m$^2$ and $T'_{in}$ is the virtual temperature of indoor air. Equation (15) is the formula of $T'_{in}$ and $T'_r$, where $e_a$ represents the actual water vapor pressure of indoor air.

$$T' = T(1 + 0.378 e_a / P) \tag{15}$$

where the absolute humidity of air saturated water under the plastic covering film $H_{s,r}$ is defined by:

$$H_{s,r} = 2165 e_a / P \tag{16}$$

In Equation (1), $\phi_e$ and $\phi_a$ are calculated as follows [41]:

$$\phi_e = \left( \frac{\eta A_r h_P}{A_{gr} \lambda} \right) Q_P \tag{17}$$

$$\phi_a = \psi_a (H_{in} - H_{out}) \tag{18}$$

Due to geographical factors, winter in Northeast China is cold and dry with strong winds. This cold wind is an important factor affecting greenhouse humidity. Therefore, cold air penetration was added to the humidity balance model. The calculation equation is shown in Equation (19):

$$\psi_a = \frac{\varepsilon V_a}{3600} \tag{19}$$

Cold air infiltration, $\psi_a$, is greatly influenced by outdoor wind speed, and $\varepsilon$ represents the cold air infiltration coefficient, the value of which is different with the outdoor wind speed and generally lies between 0.2 and 0.5.

Equations (2)–(12) were substituted into Equation (1) to obtain the temperature dynamic model of the system, which is shown in Equation (20). Equations (13)–(19) were

substituted into Equation (1) to acquire the humidity dynamic model of the system, which is shown in Equation (21). The model parameters are provided in Appendix A in Table A1.

The dynamic model of CSG shows that the model parameters, such as global radiation, outside wind speed, and outside air temperature, reflect stochastic properties. Therefore, the CSG system is a complex system characterized by uncertain parameters. Moreover, the process of CSG is a nonlinear system because dynamic models contain nonlinear items.

$$
\dot{T}_{in}(t) = \left[ -\frac{h_c A_{su}}{A_{gr}\rho C_P h} - \frac{2}{305\left(\frac{D}{v_{in}}\right)^{0.5}h} - \frac{A_h \alpha_h}{\rho C_P h} - \frac{A_w \alpha_w}{\rho C_P h} - \frac{e_s \beta}{\rho C_P h r} \right] T_{in}(t) + \frac{h_l P}{8.036\gamma\rho C_P h} H_{in}(t) +
$$
$$
\frac{12\eta}{A_{gr}\rho C_P h} u(t) + \frac{273.15}{A_{gr}\rho C_P h} + \frac{c_1 \tau S_{out} A_{gr} sinI^0}{V_a \rho c_p h} + \frac{T_l}{305(D/v_{in})^{0.5}h} + \frac{h_c t_{out} A_{su}}{A_{gr}\rho C_P h} + \frac{T_s}{305(D/v_{in})^{0.5}h}
$$
$$
+ \frac{T_h A_h \alpha_h}{\rho C_P h} + \frac{T_w A_w \alpha_w}{\rho C_P h} - \frac{C_l R_n}{\rho C_P h}
$$
(20)

$$
\dot{H}_{in}(t) = \frac{e_s \beta}{r\lambda h} T_{in}(t) - \left( \frac{h_l P}{8.036\gamma\lambda h} + \frac{\phi_a}{h} \right) H_{in}(t) - \frac{\eta A_r h_p}{A_{gr}\lambda h} u(t) + \frac{\phi_a}{h} + \frac{C_l R_n}{\lambda h}
$$
$$
+ \frac{0.00164 A_r (1+0.378e_a/P)^{1/3}}{A_{gr}h} (T_{in}(t) - T_r)^{1/3} \left[ \frac{2165e_a}{T_{in}(t)+273.15} - H_{in}(t) \right]
$$
(21)

where the system state variable can be selected as:

$$
x(t) = [x_1(t), x_2(t)]^T = [T_{in} \ H_{in}]^T
$$

The input variables are expressed as follows:

$$
u(t) = Q_P
$$

The output variables are obtained as:

$$
y(t) = \begin{bmatrix} y_1(t) & y_2(t) \end{bmatrix}^T = Cx(t) = \begin{bmatrix} 1 & 0 \\ 0 & 0 \end{bmatrix} \begin{bmatrix} x_1(t) \\ x_2(t) \end{bmatrix}
$$

## 3. Nonlinear Adaptive Control Based on Switching Mechanism

### 3.1. Controller Design Model

The dynamic model of a northern greenhouse is shown in Equations (20) and (21). The north solar greenhouse model can be re-expressed as:

$$
\dot{x}_1(t) = -a_0 x_1(t) + a_1 x_2(t) + a_2 u(t)
$$
(22)

$$
\dot{x}_2(t) = a_3 x_1(t) - a_4 x_2(t) - a_5 u(t) + a_6 (x_1(t) - 10)^{1/3} \left( \frac{8053.8}{x_1(t) + 273.15} - x_2(t) \right)
$$
(23)

where,

$$
a_0 = \frac{h_c A_{su}}{A_{gr}\rho C_P h} + \frac{2}{305h/(D/v_{in})^{0.5}} + \frac{A_h \alpha_h}{\rho C_P h} + \frac{A_w \alpha_w}{\rho C_P h} + \frac{e_s \beta}{\rho C_P h \gamma}
$$
$$
a_1 = \frac{h_l P}{8.036\gamma\rho C_P h}, \ a_2 = \frac{12\eta}{A_{gr}\rho C_P h}, \ a_3 = \frac{e_s \beta}{r\lambda h}, \ a_4 = \frac{h_l P}{8.036\gamma\lambda h} + \frac{\varepsilon V_a}{3600h},
$$
$$
a_5 = \frac{\eta A_r h_p}{A_{gr}\lambda h} \text{ and } a_6 = \frac{0.00164 A_r (1+0.378e_a/P)^{1/3}}{A_{gr}h}.
$$

In this paper, first order Taylor expansion was used to approximate the derivative of $x_1$ and $x_2$ by discarding the high order error items. The following discrete-time system can approximate the dynamic properties of the continuous greenhouse system.

$$
x_1(t+1) = x_1(t) + \dot{x}_1(t)T
$$
(24)

$$
x_2(t+1) = x_2(t) + \dot{x}_2(t)T
$$
(25)

$$
y(t) = x_1(t) + \omega(t)
$$
(26)

where $T$ is the sampling period. $x_1(t)$, $x_2(t)$ represent corresponding state variables at sampling instants of the continuous system. $\omega(t)$ is the measurement noise.

Substituting Equation (22) into Equation (24), we obtain:

$$
\begin{aligned}
x_2(t) &= \frac{x_1(t+1)-x_1(t)}{a_1 T} + \frac{a_0 x_1(t) - a_2 u(t)}{a_1} \\
&\triangleq f_1(y(t+1), y(t), u(t))
\end{aligned}
\tag{27}
$$

Thus,

$$
\begin{aligned}
x_2(t-1) &= \frac{x_1(t)-x_1(t-1)}{a_1 T} + \frac{a_0 x_1(t-1) - a_2 u(t-1)}{a_1} \\
&\triangleq f_1(y(t), y(t-1), u(t-1))
\end{aligned}
\tag{28}
$$

Furthermore, noticing Equations (23), (25) and (28), we obtain:

$$
\begin{aligned}
x_2(t) &= x_2(t-1) + \dot{x}_2(t-1)T = f_1(x_1(t), x_1(t-1), u(t-1)) + \{a_3 x_1(t-1) \\
&\quad -a_4 x_2(t-1) - a_5 u(t-1) - a_6(x_1(t-1)-10)^{1/3}\left[\frac{8053.8}{x_1(t-1)+273.15} - x_2(t-1)\right]\}T \\
&\triangleq f_2(y(t), y(t-1), u(t-1))
\end{aligned}
\tag{29}
$$

Noticing Equations (24) and (26), we obtain:

$$
\begin{aligned}
y(t+1) &= x_1(t+1) = x_1(t) + \dot{x}_1(t)T = x_1(t) + (-a_0 x_1(t) \\
&\quad +a_1 f_2(y(t), y(t-1), u(t-1)) + a_2 u(t))T \\
&\triangleq f_3(y(t), y(t-1), u(t), u(t-1))
\end{aligned}
\tag{30}
$$

Applying a similar approach in [42], the greenhouse dynamical model can be decomposed into a linear model incorporating a nonlinear term nearby the operating point, which can be expressed in the following formulation:

$$
A\left(z^{-1}\right)y(t) = B\left(z^{-1}\right)u(t-1) + v(t-1)
\tag{31}
$$

where $A\left(z^{-1}\right) = 1 + az^{-1}$, $B\left(z^{-1}\right) = 1 + bz^{-1}$. $a$ and $b$ are polynomials about $z^{-1}$. $n_a$ and $n_b$ are the system orders. $v[x(t)] = v[y(t), \ldots, y(t-n_a+1), u(t), \ldots, u(t-n_b)]$ is the higher order nonlinear item.

### 3.2. Nonlinear Controller

Before introducing the nonlinear controller, a linear controller was first introduced based on generalized minimum variance. The control object can be obtained as follows:

$$
A\left(z^{-1}\right)y(t) = B\left(z^{-1}\right)u(t-1) + \omega(t-1)
\tag{32}
$$

The generalized minimum variance performance index was introduced in the following function:

$$
J = \left\| y(t+1) - M\left(z^{-1}\right)w(t) + Q\left(z^{-1}\right)u(t)\right\|^2
\tag{33}
$$

where $M\left(z^{-1}\right)$ and $Q\left(z^{-1}\right)$ are polynomial about $z^{-1}$.

The Diophantine equation was introduced as follows:

$$
1 = FA\left(z^{-1}\right) + z^{-1}D\left(z^{-1}\right)
\tag{34}
$$

where $F$ is a constant. $D\left(z^{-1}\right)$ is diagonal polynomial matrix.

Substituting Equation (34) into Equation (32) yields:

$$
y(t+1) = D\left(z^{-1}\right)y(t) + FB\left(z^{-1}\right)u(t) + F\omega(t)
\tag{35}
$$

Substituting Equation (35) into Equation (33), the generalized minimum variance linear controller can be expressed as follows:

$$\left[FB\left(z^{-1}\right) + Q\left(z^{-1}\right)\right]u(t) = M\left(z^{-1}\right)w(t) - D\left(z^{-1}\right)y(t) - F\omega(t) \qquad (36)$$

where $H(z^{-1}) = FB(z^{-1}) + Q(z^{-1})$. The linear feedback controller based on generalized minimum variance is shown in Figure 4.

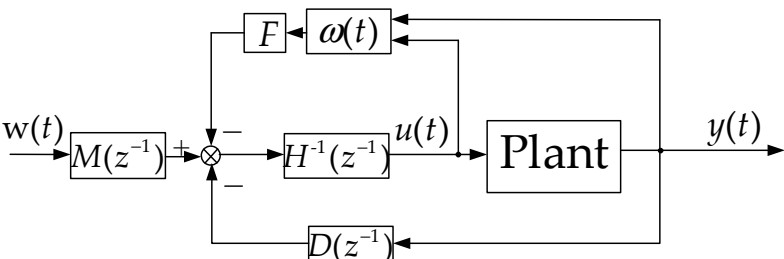

**Figure 4.** Linear feedback controller based on generalized minimum variance.

However, the linear feedback controller could not meet the actual control requirements when applying it to the complex nonlinear system. In order to control complex dynamic systems better, a nonlinear controller based on generalized minimum variance was constructed in this paper.

In order to effectively control the nonlinear plant (31), the nonlinear controller can be expressed as follows:

$$u(t) = \overline{L}^{-1}\left(z^{-1}\right)\left[\overline{M}\left(z^{-1}\right)w(t) - \overline{D}\left(z^{-1}\right)y(t) - \overline{K}\left(z^{-1}\right)v(t)\right] \qquad (37)$$

where $\overline{M}(z^{-1})$, $\overline{L}(z^{-1})$ and $\overline{D}(z^{-1})$ are diagonal polynomial matrices. $\overline{K}(z^{-1})$, which is a polynomial matrix about $z^{-1}$, is used to eliminate the effect of the nonlinear term $v(t)$, $\overline{L}(z^{-1}) = 1 - z^{-1}$. $w(t)$ is defined as bounded reference input.

Substituting Equation (37) into Equation (31) yields:

$$\left[\overline{L}\left(z^{-1}\right)A\left(z^{-1}\right) + z^{-1}B\left(z^{-1}\right)\overline{D}\left(z^{-1}\right)\right]y(t+1) = B\left(z^{-1}\right)\overline{M}\left(z^{-1}\right)w(t) + \left[\overline{L}\left(z^{-1}\right) - B\left(z^{-1}\right)\overline{K}\left(z^{-1}\right)\right]v(t) \qquad (38)$$

where $\left[\overline{L}(z^{-1})A(z^{-1}) + z^{-1}B(z^{-1})\overline{D}(z^{-1})\right]$, $B(z^{-1})\overline{M}(z^{-1})$, and $\overline{\left[\overline{L}(z^{-1}) - B(z^{-1})\overline{K}(z^{-1})\right]}$ are diagonal matrices.

The influence of $\left[\overline{L}(z^{-1}) - B(z^{-1})\overline{K}(z^{-1})\right]v(t)$ can be removed by making an adequate selection of $\overline{K}(z^{-1})$, which implies the influence of nonlinear term $v(t)$.

According to Equation (38), the closed-loop characteristic polynomial of the system is as follows:

$$T\left(z^{-1}\right) = \overline{L}\left(z^{-1}\right)A\left(z^{-1}\right) + z^{-1}B\left(z^{-1}\right)\overline{D}\left(z^{-1}\right) \qquad (39)$$

According to Equation (38), in order to eliminate the effect of nonlinear term, $\overline{K}(z^{-1})$ was chosen to satisfy the following Equation:

$$\overline{L}\left(z^{-1}\right) = B\left(z^{-1}\right)\overline{K}\left(z^{-1}\right) \qquad (40)$$

The nonlinear control strategy can be seen in Figure 5.

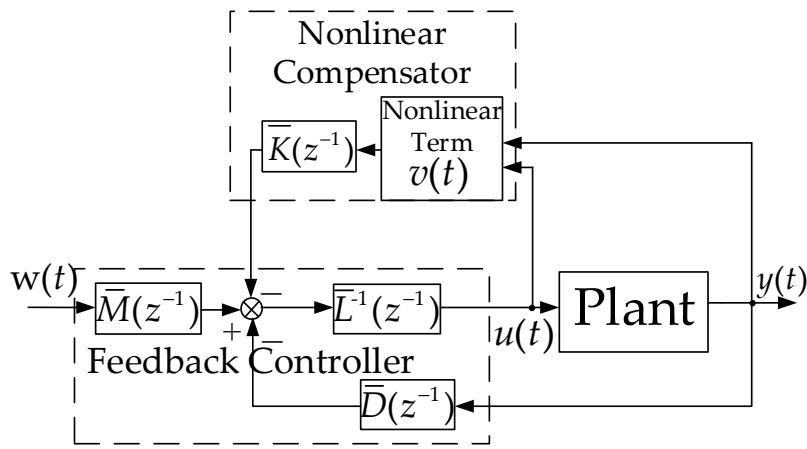

**Figure 5.** Nonlinear control strategy.

*3.3. Parameters Selection*

In order to identify parameters of the controller (37), the following cost function was introduced:

$$\psi = \|y(t+1) - M\left(z^{-1}\right)w(t) + Q\left(z^{-1}\right)u(t) + K\left(z^{-1}\right)v(t)\|^2 \tag{41}$$

where $M\left(z^{-1}\right)$, $Q\left(z^{-1}\right)$, $K\left(z^{-1}\right)$ are polynomial about $z^{-1}$.

The following Diophantine equation was introduced:

$$1 = FA\left(z^{-1}\right) + z^{-1}D\left(z^{-1}\right) \tag{42}$$

Substituting Equation (42) into Equation (31) yields:

$$y(t+1) = D\left(z^{-1}\right)y(t) + FB\left(z^{-1}\right)u(t) + Fv(t) \tag{43}$$

Substituting Equation (43) into Equation (41), if the performance index is $\psi = 0$, the generalized minimum variance controller can be obtained as follows:

$$\left[FB\left(z^{-1}\right) + Q\left(z^{-1}\right)\right]u(t) = M\left(z^{-1}\right)w(t) - D\left(z^{-1}\right)y(t) - \left[F + K\left(z^{-1}\right)\right]v(t) \tag{44}$$

A constant $\lambda$ is introduced, we selected $\lambda$ which satisfied the performance index of Equation (41).

$$M\left(z^{-1}\right) = \lambda^{-1}\overline{M}\left(z^{-1}\right) \tag{45}$$

$$Q\left(z^{-1}\right) = \lambda^{-1}\overline{L}\left(z^{-1}\right) - FB\left(z^{-1}\right) \tag{46}$$

$$K\left(z^{-1}\right) = \lambda^{-1}\overline{L}\left(z^{-1}\right)B^{-1}\left(z^{-1}\right) - F \tag{47}$$

$$\det\left\{\overline{L}\left(z^{-1}\right)A\left(z^{-1}\right) + z^{-1}\lambda B\left(z^{-1}\right)D\left(z^{-1}\right)\right\} \neq 0, |z| < 1 \tag{48}$$

*3.4. Adaptive Switching Control*

The parameters of the greenhouse model always vary as the external environment changes. These situations directly lead to the occurrence of parameter uncertainties. Therefore, it is necessary to update model parameters of CSG in real time. According to Equation (37), the following formula is given to identify the system parameters: $y(t) = \theta^T X(t-1) + v(t-1)$, where $X(t-1) = [y(t-1), \ldots, y(t-n_a), u(t-1), u(t-n_b-1)]$, $\theta = \left[-a_1, \ldots, -a_n, b_0, \ldots, b_{n_b}\right]$.

To predict the output of the system, two estimation models were introduced in this paper. The first one was the linear estimation model:

$$\hat{y}_1(t) = \hat{\theta}_1^T(t-1)X(t-1) \tag{49}$$

where $\theta$ can be approximated as $\hat{\theta}_1^T(t-1)$ at instant $t-1$ and the parameter $\theta$ can be determined through the algorithm as follows:

$$\hat{\theta}_1(t) = \hat{\theta}_1(t-1) + \frac{\mu_1(t)X(t-1)e_1^T(t)}{1 + X(t-1)^T X(t-1)} \tag{50}$$

$$\mu_1(t) = \begin{cases} 1 & if\,\|e_1(t)\| > 4\triangle \\ 0 & else \end{cases} \tag{51}$$

where $\triangle > 0$ is the upper bound of the nonlinear term $v(t-1)$. The linear model error $e_1(t)$ is defined as follows:

$$e_1(t) = y(t) - \hat{y}_1(t) = y(t) - \hat{\theta}_1^T(t-1)X(t-1) \tag{52}$$

The neural network nonlinear estimation model, written in the following algorithm, is the second model.

$$\hat{y}_2(t) = \hat{\theta}_2^T(t-1)X(t-1) + \hat{v}(t-1) \tag{53}$$

where $\hat{v}(t-1)$ is approximated through RBF neural networks and $\hat{\theta}_2^T(t-1)$ is the second approximation of $\theta$ at instant $t-1$. The algorithm was employed to identify the parameter as follows:

$$\hat{\theta}_2(t) = \hat{\theta}_2(t-1) + \frac{\beta(t)X(t-1)e_2^T(t)}{1 + X(t-1)^T X(t-1)} \tag{54}$$

$$\beta(t) = \begin{cases} 1 & if\,\|e_2(t)\| > 4\xi \\ 0 & else \end{cases} \tag{55}$$

where $|v(t-1) - \hat{v}(t-1) \le \xi|, \xi < 0$ is a pre-specified small positive number and $e_2(t)$ is the nonlinear model error, i.e.:

$$e_2(t) = y(t) - \hat{y}_2(t) = y(t) - \hat{\theta}_2^T(t-1)X(t-1) - \hat{v}(t-1) \tag{56}$$

Without considering nonlinear term $\hat{v}(t-1)$, the linear adaptive control law based on the linear estimation model can be expressed in the following Equation:

$$\overline{L}\left(z^{-1}\right)u(t) = \hat{\overline{M}}_1^t w(t) - \hat{\overline{D}}_1^t y(t) \tag{57}$$

From Equation (31), the nonlinear adaptive control law based on the RBF neural network nonlinear estimation model is expressed as follows and the structure is shown in Figure 6.

$$\overline{L}\left(z^{-1}\right)u(t) = \hat{\overline{M}}_2^t\left(z^{-1}\right)w(t) - \hat{\overline{D}}_2^t\left(z^{-1}\right)y(t) - \hat{\overline{K}}_2^t\left(z^{-1}\right)\hat{v}(t) \tag{58}$$

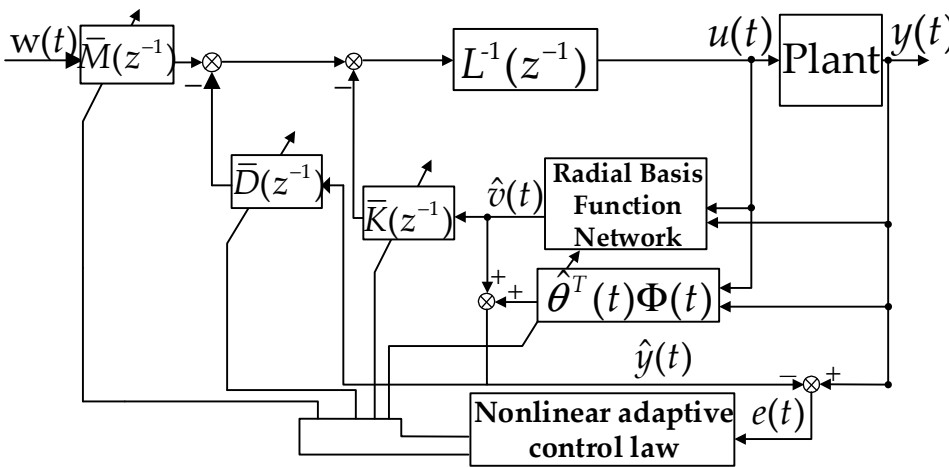

**Figure 6.** Nonlinear adaptive controller structure.

To ensure the stability of the close-loop system, the linear adaptive controller was adopted in this paper. However, the strong nonlinear system showed poor performance if a nonlinear item $\hat{v}(t-1)$ was larger when using a linear adaptive controller alone. The nonlinear adaptive controller can decrease the effect of a nonlinear term on system output. However, the nonlinear adaptive controller has an aggressive control effect and stability of the close-loop system cannot be guaranteed. In this paper, a switching mechanism was introduced to enhance performance of the control system and guarantee stability for the closed-loop system, which is shown in Figure 7. The switching criterion is defined as:

$$J_i(t) = \sum_{l=1}^{t} \frac{\mu_i(l)\left(\|e_i(l)\|^2 - 16\Delta^2\right)}{4\left(1 + X(l-1)^T X(l-1)\right)} + \alpha \sum_{l=t-N+1}^{t}(1 - \mu_i(l))\|e_i(l)\|^2 \qquad (i = 1.2) \tag{59}$$

$$\mu_i(t) = \begin{cases} 1 & if\,\|e_i(t)\| > 4\Delta \\ 0 & else \end{cases} \tag{60}$$

where $N$ is an integer and $\alpha \geq 0$ is a predefined constant, $i = 1$ denotes the linear model, and $i = 2$ stands for the nonlinear models. Each time, $t$, the nonlinear estimation model, and the linear model predicted the system output, and model parameters were automatically updated by the identified algorithm. Meanwhile, $J_1(t)$ and $J_2(t)$ were calculated respectively and the control law $u^*(t)$ was selected corresponding to the smaller $J^*(t)$ to control the system.

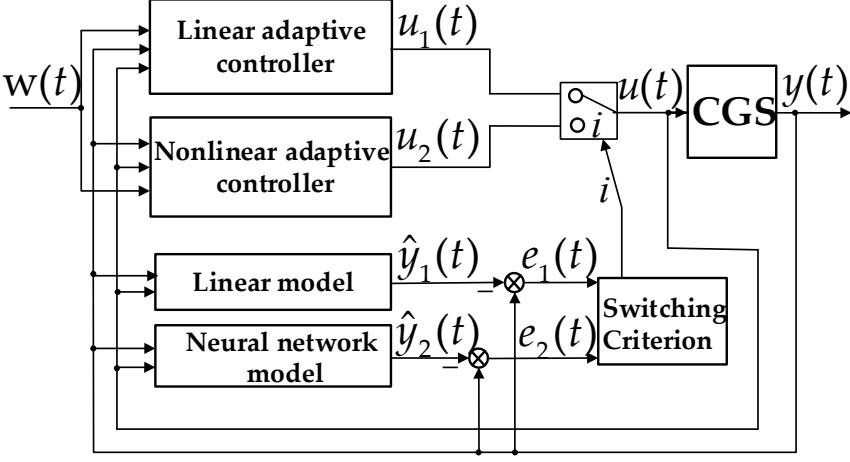

**Figure 7.** Switching mechanism based on multiple model.

### 3.5. RBF Neural Network for Unmodeled Dynamics

As we all know, neural networks are capable of approaching a complicated nonlinear mapping sufficiently owing to the rich connections and nonlinear activating functions of neurons [43]. In this research, an RBF network was employed to estimate and compensate the unmodeled dynamics of the CSG system.

The RBF neural network was divided into three layers herein in this paper, the input layer which connects the input vector to the network, the unique hidden layer, and the output layer. The activation function of RBF neural network was the Gaussian function, which can be expressed as [44]:

$$F(x) = \exp\left(\frac{\|x - c\|^2}{2\sigma^2}\right) \tag{61}$$

where $x$ is the $n$-dimensional input vector and $c$ is the center vector, which is the same as the $x$-dimension, and $\sigma$ is the width of the basis function around the center point.

In this paper, the output of the neural network input layer was $\hat{v}_i[x(t)]$. The input vector was $x(t) = [y(t), \ldots, y(t - n_a + 1), u(t), \ldots, u(t - n_b)]$, and the output layer network nonlinear output was as follows:

$$\hat{v}_i[x(t)] = \sum_{p=1}^{q} W_{pm} F_p(x) \tag{62}$$

where, $m = 1, 2, \ldots, l$ and $q$ is the number of nodes in the hidden layer, $l$ is the number of nodes in the output layer, $W_{pm}$ is the connection weight between the neuron $P$ in the hidden layer and the neuron m in the output layer, and $F_p(x)$ is the excitation function of the neuron $P$ in the hidden layer.

## 4. Results

In this part, two simulation experiments, being composed of a set-point tracking experiment and a full-day real weather experiment, were conducted to test the performance of the nonlinear adaptive control strategy.

### 4.1. Set-Point Tracking Experiment

The research was designed to prove the effectiveness of the control method for CSG in terms of the tracking performance with strong multi-disturbances. There existed internal conditions as follows: solar radiation $S_{out}$ = 350 W/m$^2$, outside air temperature $t_{out}$ = 5 °C, outside humidity $H_{out}$ = 16 g/m$^3$, outside wind speed $v_{out}$ = 2 m/s, inside temperature $t_{in}$ = 17 °C, and inside humidity $H_{in}$ = 18 g/m$^3$. After using the Euler method, the initial design models were expressed around the nominal operating point as follows: $A(z^{-1}) = 1 - 1.992z^{-1} + 0.9851z^{-2}$, $B(z^{-1}) = 0.004321 - 0.4223z^{-1}$, where the system order was $n_a = 2$ and $n_b = 1$. In this situation $\lambda$ was chosen as 0.2 and the switching criterion parameters were selected to be $\alpha$ = 1, $N$ = 2, and $\Delta$ = 0.015. The initial weights of the RBF neural network were obtained by training the input and output data in a small range of working points. The hidden layer was equal to 8 and the relevant parameters were chosen to be $q = 6$, $\sigma = 0.65$, $\alpha = 0.05$, and $\eta_{RBF} = 0.3$.

In order to research the tracking performance of the nonlinear adaptive controller, the inside temperature set points were changed to a wide range. At the same time, the outside weather conditions, such as outside temperature, outside wind speed, and outside solar radiation fluctuated in a large scope. The experiment design was as follows. The inside temperature was changed from 17 °C to 28 °C at t = 0–500 s. The effects of the external disturbances were simulated in this process. The outside temperature was changed from 5 °C to −3 °C at t = 150 s. The solar radiation was changed from 350 W/m$^2$ to 150 W/m$^2$ at t = 350 s. The inside temperature was changed from 28 °C to 24 °C at t = 500 s and the outside solar radiation simultaneously became zero. The outside wind speed was changed

from 2 m/s to 6 m/s at t = 700 s and the outside temperature was changed from −3 °C to −12 °C at t = 850 s. In the end, the inside temperature was changed from 24 °C to 19 °C at t = 1000 s. The effect of the extreme outside temperature was simulated during this period. The outside temperature was changed from −16 °C to −27 °C at t = 1200 s.

　　The set-point tracking research results are demonstrated in Figures 8 and 9. The inside temperature quickly tracked the set point and the control method reduced the influence of uncertain factors. Moreover, in the face of strong external disturbances such as stiff wind and cold weather, the inside temperature still tracked the set point quickly.

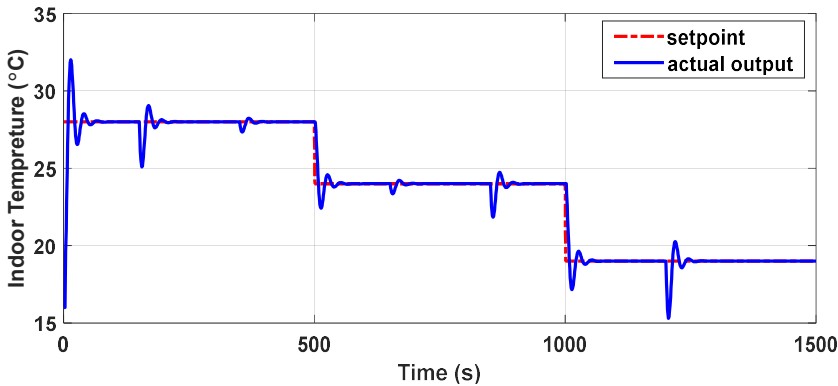

**Figure 8.** Response of inside temperature.

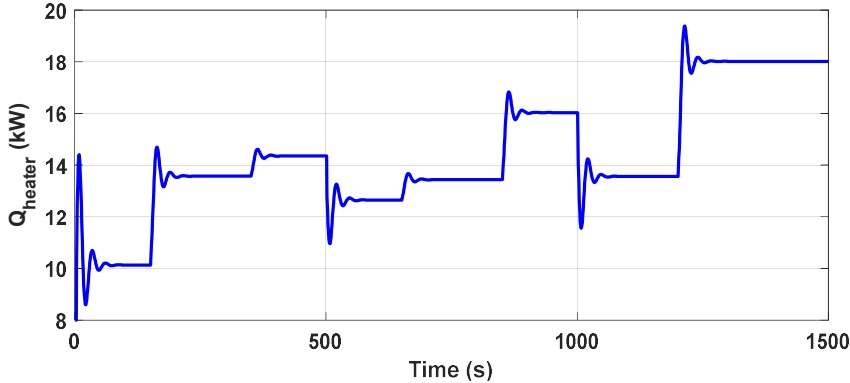

**Figure 9.** Corresponding control input.

### 4.2. Full-Day Real Weather Experiment

　　In this test, we used experimental data from a full, cold day (26 January 2018) in Shenyang, China. The environmental conditions (including inside temperature, outside wind, outside humidity, outside temperature, and solar radiation) from 0:00 to 24:00 were collected every 15 min. Figures 10–14 express the inside temperature, wind speed, outside humidity, outside temperature and solar radiation, respectively. It can be derived from Figures 10–14 that the aforementioned model parameters show random properties. Figure 10 shows that the inside temperature exceeded 15 °C from 9:30 to 16:00 on 26 January 2018, and at other times was below 15 °C, which was unfavorable for crop growth. We determined the set point of the inside temperature according to the following rules. Firstly, the set point was designed according to the current temperature of the greenhouse. The air heater equipment shut off when the current temperature exceeded 20 °C. Secondly, the set point of the temperature was adjusted by the energy management. The set point gradually decreased to 15 °C with the reduction of outside solar radiation, which met the minimum requirement for the crops. The set point curve of the temperature is shown in Figure 10 (red line). The air heater equipment was turned on and off at 15:30 and 10:00, respectively. The outside weather data were substituted into the dynamical model of the

CSG during the experiment. The corresponding response for the set point of the inside temperature is shown in Figure 15. The corresponding response of control input is shown in Figure 16. The tracking error curves, corresponding to the conventional PID method and the presented nonlinear adaptive scheme, are shown in Figures 17 and 18.

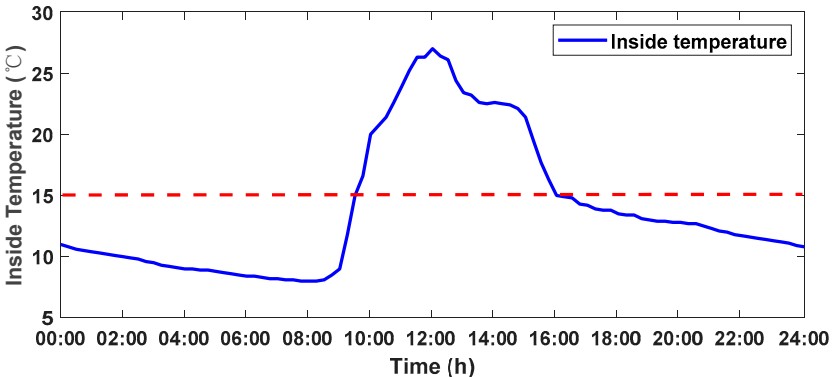

**Figure 10.** The inside temperature of the CSG on 26 January 2018.

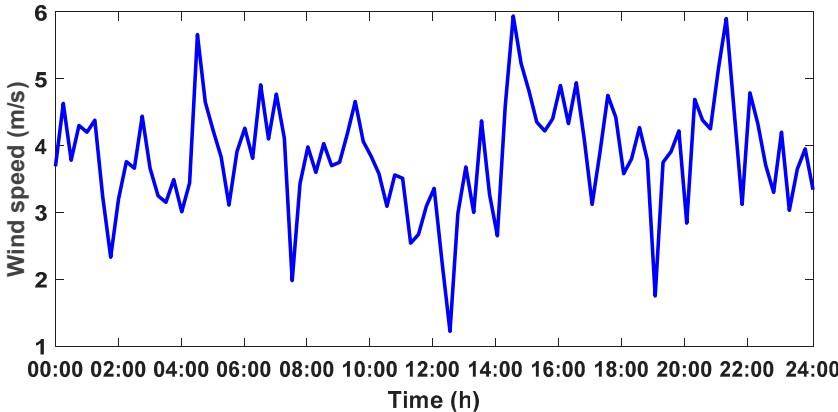

**Figure 11.** The outside wind speed of the CSG on 26 January 2018.

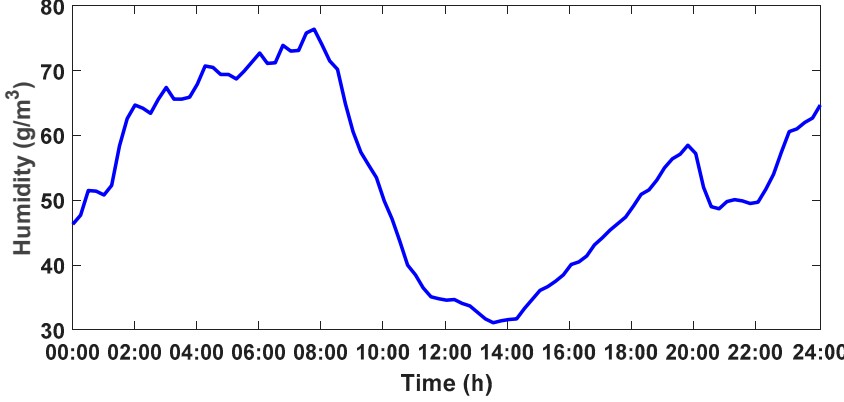

**Figure 12.** The outside humidity of the CSG on 26 January 2018.

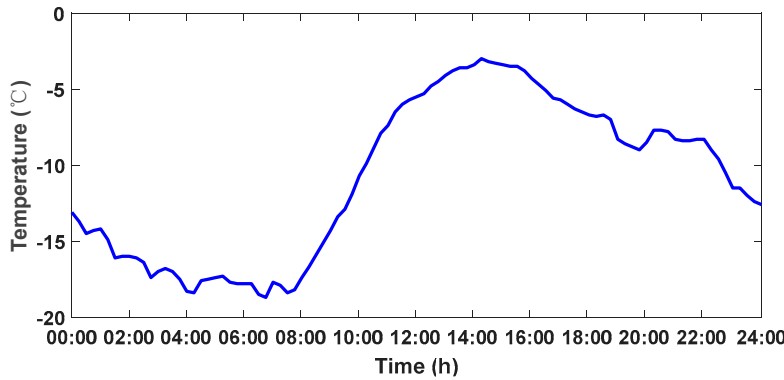

**Figure 13.** The outside temperature of the CSG on 26 January 2018.

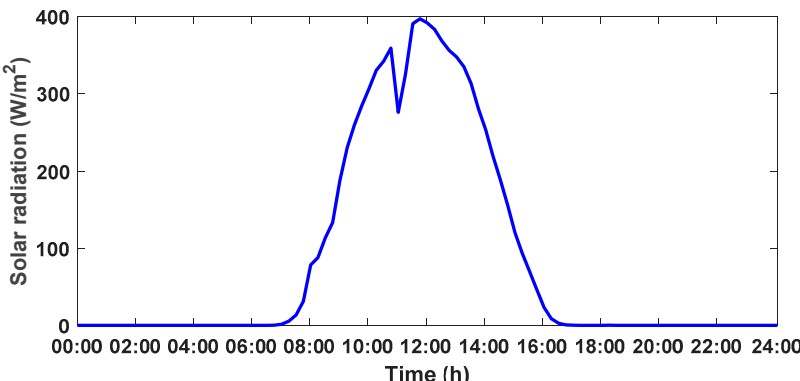

**Figure 14.** The outside solar radiation of the CSG on 26 January 2018.

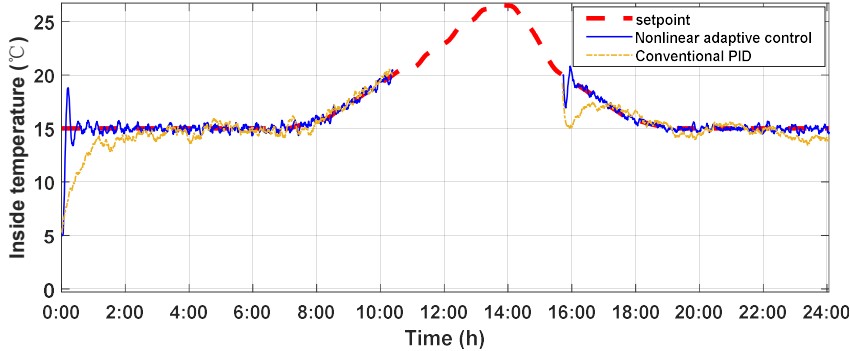

**Figure 15.** The corresponding response for set point of inside temperature.

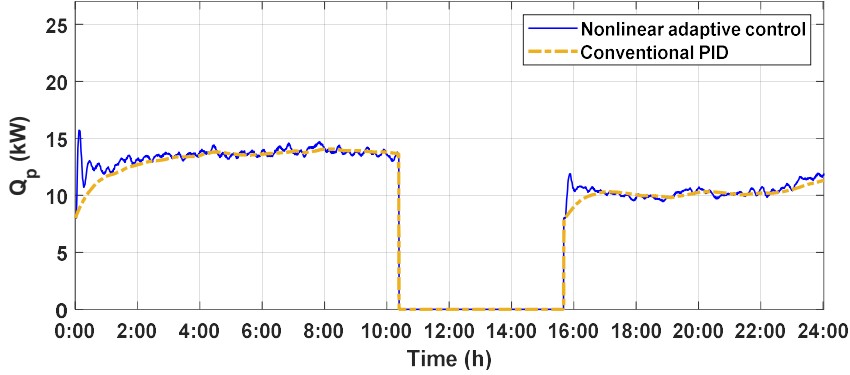

**Figure 16.** The corresponding response of control input.

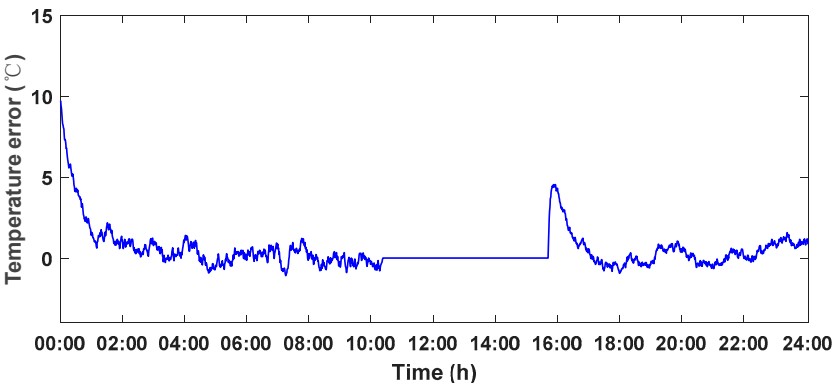

**Figure 17.** Variation errors of conventional PID control during the experimental process.

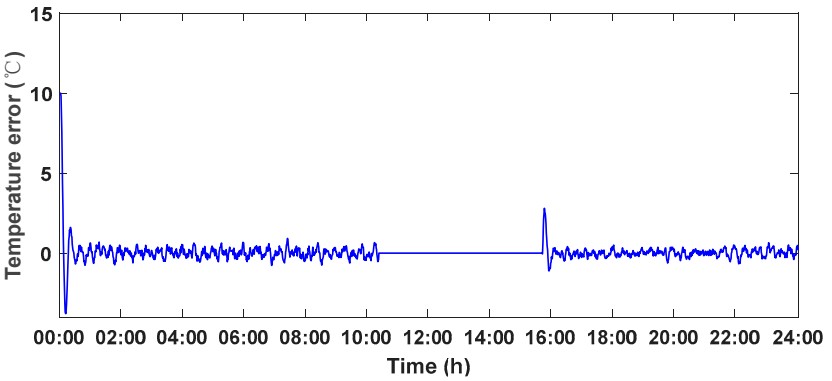

**Figure 18.** Variation errors of nonlinear adaptive control errors during the experimental process.

As indexes of the variable randomness, average error and standard error were selected to measure the control performance of using a conventional PID method and an RBF nonlinear adaptive scheme. Table 1 shows the consequence of using the conventional PID method and the presented nonlinear adaptive strategy. The mean errors were 0.8460 and 0.2967, respectively. The standard errors were 1.8480 and 1.3342, respectively. Moreover, as can be seen from Figures 17 and 18, the tracking error of temperature for the presented nonlinear control scheme was smaller than the error of the conventional PID method. By comparing the two methods, it can be seen that the presented nonlinear adaptive control scheme showed better control performance, characterized by good adaptability and preferable robustness. Compared to the conventional PID method, the proposed nonlinear adaptive scheme has the following advantages. Firstly, the strategy improves set-point tracking performance greatly and is more robust. Secondly, it has a better adaptability for external climatic disturbance. Thirdly, it may give a meaningful reference to deal with the complex greenhouse climate control problem.

**Table 1.** Performance Comparison between conventional PID control and nonlinear RBF adaptive control.

| Methods | Temperature Error (°C) | | Corresponding Line Front |
| :---: | :---: | :---: | :---: |
| | **Mean** | **Standard** | |
| Conventional PID | 0.8460 | 1.8480 | orange, dash-dot |
| Nonlinear adaptive control | 0.2967 | 1.3342 | blue, solid |

## 5. Conclusions

In this paper, the CSG has been described as a nonlinear, uncertain and multi-disturbance dynamic system. A nonlinear dynamic model for CSGs, based on energy

conservation laws, was constructed by equations, and the corresponding control model was established. A nonlinear adaptive control strategy for CSG production, employing an RBF neural network, was proposed. The main objective was to meet the normal requirements of temperature control for CSGs. Due to the great ability to deal with a non-minimum phase system, a generalized minimum variance method was introduced to determine the controllers' parameters. Considering the strong learning capacity of the RBF neural network, the RBF neural network was employed to estimate and compensate the unmodeled dynamics of the system. The mean error and standard error for the conventional PID method were 0.8460 and 1.8480, respectively. By contrast, the presented nonlinear control strategy had great improvement with the result of 0.2967 and 1.3342, corresponding to the mean and standard error. The control strategy was tested for complex greenhouse climate control and the experiment results showed that the presented nonlinear adaptive control method had great adaptability, robustness, and prominent real-time control performance. A valuable reference can be provided by this control method to formulate climate control schemes for practical application in greenhouse production.

**Author Contributions:** Y.W., Y.L. and R.X. conceived and designed the experiments; Y.W. and Y.L. performed the experiments; Y.W. and Y.L. analyzed the data; Y.W. and Y.L. supervised the experiment; Y.W. and Y.L. wrote the manuscript; Y.W., Y.L. and R.X. reviewed the manuscript. All authors have read and agreed to the published version of the manuscript.

**Funding:** This project was supported by the National Natural Science Foundation of China (NSFC) (61673281,32001415,61903264) and the Natural Science Foundation of Liaoning Province (2019-KF-03-01).

**Data Availability Statement:** Data is contained within the article.

**Acknowledgments:** The authors sincerely thank the National Natural Science Foundation of China for their financial support. We gratefully acknowledge the assistance of Tan Liu, Qingyun Yuan, Dapeng Zhang, Nannan Zhang in revised manuscript. We are also grateful to reviewers for their recommendations to improve the quality of this paper.

**Conflicts of Interest:** The authors declare no conflict of interest.

## Appendix A

**Table A1.** Meanings of parameters of the greenhouse temperature model.

| Parameter | Meaning | Value Range | Unit |
|---|---|---|---|
| P | standard atmospheric pressure | 101 | kPa |
| $\rho$ | air density | 1.1691 | $kg/m^3$ |
| $C_P$ | the specific heat of air at constant pressure | 1.003 | - |
| $e_s$ | the air saturation vapor pressure | 3.167 | kPa |
| $\gamma$ | the psychrometric constant | 66 | Pa/$^\circ$C |
| $\lambda$ | latent heat of evaporation | 2450 | J/g |
| $v_{out}$ | outside wind speed | 0.2–12 | m/s |
| $C_l$ | the coefficient of convective heat loss from indoor air to the cover | (0.05–50) | - |
| $\eta$ | heat energy efficiency of the heating equipment | 0.85 | - |
| $R_n$ | the net solar radiation absorbed by crops | 100–350 | $W/m^2$ |
| $\beta$ | influence coefficient of temperature change on saturated water vapor pressure | 0.001 | - |
| $H_{out}$ | outside humidity | 6–29 | $g/m^3$ |
| $A_w$ | north wall area | 50 | $m^2$ |
| $t_w$ | north wall temperature | 8–20 | $^\circ$C |
| $\alpha_w$ | convective heat transfer coefficient through north wall | 5–25 | - |
| $h_l$ | the heat transfer constant between crops and inside air | 13.3 | - |
| D | leaf width | 0.15–0.25 | m |

**Table A1.** *Cont.*

| Parameter | Meaning | Value Range | Unit |
|---|---|---|---|
| $\varepsilon$ | cold air permeability coefficient | 0.2–0.5 | m/s |
| $\tau$ | the greenhouse global transmission | 0.6 | - |
| $A_{gr}$ | surface area which absorbs solar radiation | 392 | m$^2$ |
| $A_{su}$ | the superficial area of the cover materials | 615 | m$^2$ |
| $h$ | the height of greenhouse | 2.5 | m |
| $A_h$ | the area of north roof | 100 | m$^2$ |
| $t_h$ | north roof temperature | 8–25 | °C |
| $c_1$ | the aging coefficient of lighting material | 0.82 | - |
| $S_{out}$ | solar radiation | 100–500 | W/m$^2$ |
| $v_{in}$ | inside wind speed | 0–0.3 | m/s |
| $t_s$ | soil surface temperature | 6–25 | °C |
| $t_l$ | the leaf temperature of crops | 6–20 | °C |
| $\alpha_h$ | convective heat transfer coefficient through north roof | 5–25 | - |
| $t_{out}$ | outside air temperature | −30–8 | °C |

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
