# Peer review of "Application of Nonlinear Adaptive Control in Temperature of Chinese Solar Greenhouses"

_electronics, doi:10.3390/electronics10131582_

Round 1
Reviewer 1 Report
- Interesting research in a well drafted manuscript that needs some small improvements.
- Abstract is okay but is not likely to entice the readership to continue reading the rest of the manuscript.
- Use of acronyms in an abstract is unlikely to attract readers not already aware of the manuscript’s content.
- Results are only presented in weak, qualitative fashion. Highest quality expression of main conclusions or interpretations is quantitative results discussed in the broadest context possible, e.g., percent performance improvement compared to a declared benchmark. “…can achieve excellent performance…” is very weakly stated results compared to “…xxx percent performance improvement over conventional methods was achieved….”
- Avoid using the currently popular superfluously added salutation “actually” as done in line 10 (and again later in line 225). The usage implies the statement is “actually” true unlike other claims made. In this instance the sentence implies greenhouse systems are not actually complex (despite the claim), but conventional control methods actually struggle, nonetheless.
- Introduction is decently done with some omitted literature and some mild abuse of multi-citation without elaboration.
- Literature review well documents the lineage of the feedback instantiations of nonlinear adaptive control and some feedforward implementations including stochastic instantiations of artificial intelligence but omits deterministic artificial intelligence approaches stemming from the same lineage. Please reference deterministic artificial intelligence (e.g. https://doi.org/10.3390/jmse8080578) as a competing alternative.
- Please elaborate a reason for the reader to investigate each of the triple cited references [1-3].
- Please elaborate a reason for the reader to investigate each of the quadruple cited references [26-29].
- Equations are well done and scientifically sound, aiding repeatability of the manuscript’s proposals.
- Figures are generally well done, but some can use some improvement.
- Figure 1 is especially valuable and lends high quality to the manuscript’s content as applicable. Please spend efforts improving figure 1 as much as possible concentrating on legibility.
- Figure 3, 4,6,7,8,9,10,11,12,13,14 contain illegible internal text due to small font size. Please notice the smallest font size permissible in the manuscript template (to ensure legibility by the reader) is the figure caption which provides a conveniently proximal prototype for sizing figures.
- Line styles and sizes are identical in figures 6,7,13,14 rendering the disparate data indistinguishable when the manuscript is read in printed hardcopy (particularly in black and white) negating the value of the figures.
- Tables are effectively used to define parameters, but tables of quantitative results are merely omitted. Please add a table of quantitative results that are self-evidently available (e.g. in figure 13, 14). Means and standard deviations of differences are ubiquitously understandable figures of merit that can be succinctly summarized in statements added to the abstract and Conclusions.
- Inclusion of the appendix is welcome and effective.
Reviewer 2 Report
The title is puzzling - is it necessary to indicate that the solar greenhouses are Chinese? What is the uniqueness of such an object? Or, are Russian, American, and Chinese greenhouses still the same type of plant? Is it possible to apply heat and moisture balance models for "Russian" greenhouses? Can the proposed control methods be applied to "Canadian" greenhouses? The manuscript does not answer these questions.
The title of the manuscript also implies a guess that known methods will be applied to a specific control object. Then it remains to be expected that the novelty of the work is associated with a new approach to the description of the plant, but unfortunately, this is not the case.
The abstract reflects the content of the manuscript quite well. The introduction, in my opinion, does not give an idea of the necessity of the problem being solved. The introduction does not open up the gap in the known researches.
Section 2 presents known equations. The model does not take into account the spatial distribution of temperature and humidity in the greenhouse space. Also, the number of sensors and their location is not discussed, which is quite important if the problem of maintaining a given temperature under sharp external changes and disturbances is considered. I think that Section 2.2 can be greatly shortened and ended with a plant model in the form of equations (22) and (23).
In sections 2 and 3, I recommend that the authors carefully check their equations and notations, as mistakes are encountered (for example, see line 175).
Section 3 is the main, but it is too mathematized and it lacks important information - why do the authors complicate the control system. In my opinion, here and in the Introduction it is necessary to clearly indicate the problems of the simplest feedback control system. Then it will be possible to understand the necessity to complicate the control system for this plant.
In general, the manuscript is devoted to the consideration of a known plant, to which well-known methods of nonlinear control are applied. In my opinion, the authors should better understand and write the contribution of their work to the field.
Round 2
Reviewer 2 Report
I thank the authors for their attentiveness to my comments. In my opinion, the main problems have been fixed.
However, the following is not entirely clear.
In line 231, the authors use vector x, and their components x_1=T_in and x_2=H_in. Besides they use designations: u=Q_p and y=T_in. So, y is a scalar. And here is a contradiction and different designations for T_in or y=x_1.
In line 218, the authors introduce ?_in=?_1, ?_in= ?_2, ?_p= ?. This is another puzzling thing. The reader would expect that y is a vector with components y_1 and y_2.
If we eliminate all the contradictions, then the question remains - why are different designations for the same values used in different sections? It will be more convenient for the reader if the authors do not unnecessarily increase the number of variables.
